# LenSiam: Self-Supervised Learning on Strong Gravitational Lens Images

**Po-Wen Chang**[*]
Ohio State University
Columbus, OH, USA
chang.1750@osu.edu

**Kuan-Wei Huang**[*]
Carnegie Mellon University
Pittsburgh, PA, USA
kuanweih@alumni.cmu.edu

**Joshua Fagin**
City University of New York
New York, NY, USA
jfagin@gradcenter.cuny.edu

**James Hung-Hsu Chan**
American Museum of Natural History
New York, NY, USA
jchan@amnh.org

**Joshua Yao-Yu Lin**[*]
University of Illinois at Urbana Champaign
Champaign, IL, USA
yaoyuyl2@illinoiis.edu

## Abstract

Self-supervised learning has been known for learning good representations from data without the need for annotated labels. We explore the simple siamese (Sim-Siam) architecture for representation learning on strong gravitational lens images. Commonly used image augmentations tend to change lens properties; for example, zoom-in would affect the Einstein radius. To create image pairs representing the same underlying lens model, we introduce a lens augmentation method to preserve lens properties by fixing the lens model while varying the source galaxies. Our research demonstrates this lens augmentation works well with SimSiam for learning the lens image representation without labels, so we name it LenSiam. We also show that a pre-trained LenSiam model can benefit downstream tasks. We open-source our code and datasets at https://github.com/kuanweih/LenSiam.

## 1 Introduction

Strong gravitational lensing is a phenomenon predicted by Einstein's theory of general relativity, in which the gravitational field of a massive foreground (e.g., galaxy or galaxy cluster with dark matter) can bend and distort the path of light from a background source. Observationally the background source can be seen as multiple images, arcs, or even rings around the foreground lensing object. In recent years, strong gravitational lensing has emerged as a powerful tool for studying the distribution of dark matter [1, 2, 3] or the Universe's expansion rate (e.g., Refs. [4, 5, 6]).

In recent years, machine learning (ML) has shed light on strong lensing science. For example, Refs. [7] and [8] show that convolutional neural network (CNN) based models could be used to estimate the values and the corresponding uncertainties of the parameters given the strong lens images. Refs. [2] and [3] show the possibility of using CNN to tackle dark matter substructures in simulated strong lensing. Ref. [9] shows that Vision Transformer (ViT) has the advantage for lens parameter

---

[*]equal contribution

NeurIPS 2023 AI for Science Workshop.

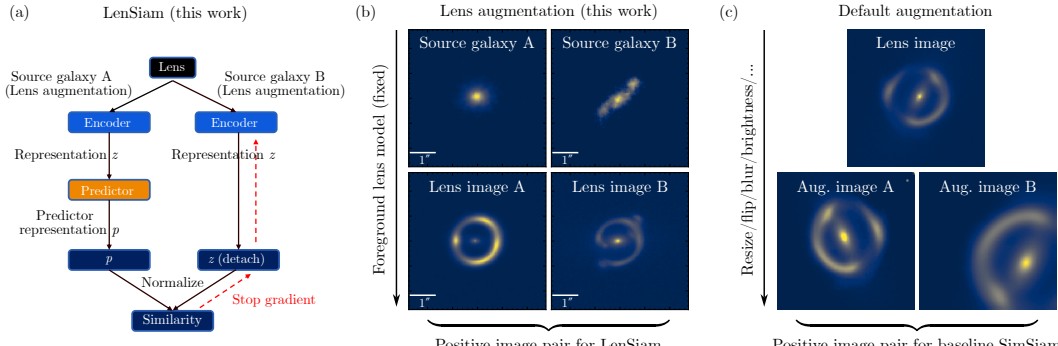

Figure 1: **(a)** The LenSiam architecture for this work. We generate positive pairs of lens images through a lens augmentation approach to learn the representation of lens images. **(b)** Example of two different COSMOS source galaxies (top) and their lens images with the identical lens model (bottom). The bottom images represent a positive lens image pair for our LenSiam models. **(c)** Example of applying the default augmentation to a lens image. The bottom augmented images represent a positive lens image pair for our baseline SimSiam models.

inference through the attention mechanism. However, most of them are trained on specific simulations which makes inference on real data across different observations challenging.

On the other hand, self-supervised learning (SSL) has emerged as a promising approach for training deep neural networks in domains where labeled data is scarce or expensive to obtain [10, 11, 12, 13, 14, 15]. By leveraging the inherent structure and patterns in the data, SSL can learn rich representations that capture the underlying structure of the data, and transfer well to downstream tasks. In astronomy, Ref. [16] shows that by learning on galaxy images, SSL could learn that strong lens images are different from galaxy images. Ref. [17] shows that SSL could be used for anomaly detection for jets in high-energy collisions. Ref. [18] uses SSL for radio galaxies classification under dataset shift.

For SSL algorithms, the SimSiam architecture [10] has been shown to be effective at learning meaningful representations of images through self-supervised training without labels. As a variant of the Siamese networks [19], SimSiam has its unique way of preventing output collapsing: the stop-gradient operation [11], and has the following characteristics amongst Siamese networks. SimSiam requires only positive image pairs (i.e., pairs of images of the same class or characteristic) during training compared to most contrastive learning methods [20] such as SimCLR [10] which repulses negative pairs while attracting positive pairs. We note that BYOL [12] relies only on positive pairs too but it uses a momentum encoder to prevent collapse. SimSiam does not need clustering [21] to avoid constant output such as SwAV [13] which incorporates online clustering.

In this paper, we present the LenSiam architecture shown in Figure 1 (a) for representation learning on strong gravitational lens image data. LenSiam combines the SimSiam architecture with a novel lens image augmentation method that is invariant for the lens model (see Section 2 for more details). We leverage the SimSiam architecture to study strong gravitational lens images as (i) gravitational lensing systems are rare and labeling lens images is traditionally difficult; (ii) we can circumvent the potential ambiguity of defining negative pairs for the nature of actual lens image observations; and (iii) SimSiam is more computationally affordable than contrastive learning methods. In Section 3, we explore both the lens image representations learned from LenSiam and a baseline SimSiam model that uses default image augmentation. We finally showcase that LenSiam does improve the performance of a downstream regression task on an independent lens image dataset.

## 2   Data and Training

In Section 2.1, we detail the strong lensing simulation for generating the datasets in this work. In Section 2.2, we describe our augmentation approach for simulated lens images. In Section 2.3, we provide the general framework to train our LenSiam and baseline SimSiam models.

## 2.1  Simulation Setup

Simulating strong gravitational lens images requires four major components: the mass distribution of the lensing galaxy, the source light distribution, the lens light distribution, and the point spread function (PSF), which convolves images depending on the atmosphere distortion and telescope structures. We use the LENSTRONOMY package [22, 23] to generate strong lens images with different combinations of lensing parameters. For the mass distribution, we adapt the commonly used [24, 25] elliptically symmetric power-law distributions [26] to model the dimensionless surface mass density of lens galaxies: $\kappa(\theta_1, \theta_2) = \frac{3-\gamma}{2} \left( \frac{\theta_E}{\sqrt{q\theta_1^2 + \theta_2^2/q}} \right)^{\gamma - 1}$ where $\theta_E$ is the circularized Einstein radius, $\gamma$ is the negative power law slope of the mass distribution (with $\gamma = 2$ corresponding to isothermal), $\theta_1$ and $\theta_2$ are the mass center coordinates, and $q$ is the minor-to-major axis ratio that is related to the ellipticities $e_1$ and $e_2$. We also include an external shear parameterized by $\gamma_1$ and $\gamma_2$. All these parameters are randomly drawn from the uniform ranges described in Ref. [27]. The light distribution of the lens galaxy and source galaxy is described by the elliptical Sérsic profile [28]. To model the instrumental effects, we convolve our lens images with 23 *Hubble Space Telescope* (*HST*) PSFs generated from Tinytim [29] by Ref. [27] and then add Gaussian white noise on them.

Our simulation uses two kinds of source galaxies. The first are real galaxy images from the COSMOS 23.5 and 25.2 data sets [30] taken from the *HST*'s COSMOS survey. We select only images with at least 50 pixels and noise root mean square deviation of less than 5%. The second are core-Sérsic profiles which is an analytic model for the brightness of elliptical galaxies. We use 75% COSMOS images and 25% analytic sources. We also add complexity to the sources by including a 10% chance for a source to be the combination of two different sources to mimic a merger of two galaxies.

Finally, each image has a dimension of $110 \times 110$ pixels with a resolution of 0.05 arcseconds per pixel and is normalized to have a maximum brightness of one. The lens light is deviated from the center-of-the-mass model and ellipticities by a Gaussian distribution with a standard deviation of 2.5% the maximum range.

## 2.2  Augmentation Methods and Datasets

To train our model with the LenSiam architecture in Figure 1 (a), we generate 100,000 positive pairs of lens images (200,000 images) with our lensing simulation pipeline and lens augmentation approach. The commonly used random augmentation methods are problematic here as the lens properties will be easily changed. For example, enlarging a lens image will directly change the Einstein radius. Therefore, for each positive image pair, we fix the foreground lens model so that the two images share the same lensing parameters $(\theta_E, e_1, e_2, \theta_1, \theta_2, \gamma, \gamma_1, \gamma_2)$, while their background sources, lens light parameters, noise properties, and PSFs are randomly varied. Figure 1 (b) shows an example of our positive lens image pair generated by two different source galaxies. This lens augmentation approach not only facilitates our model to learn a consistent representation of the underlying lens model but also allows us to take into account the diversity in the galaxy attributes of real data.

To verify the validity of our LenSiam models, we also train baseline SimSiam models by applying the default image augmentation methods of SimSiam [11] (i.e., a combination of random cropping and resize, random horizontal flip, random color distortions, and random Gaussian blur) to 100,000 randomly selected lens images. Figure 1 (c) shows that individual lens image is augmented twice to form the positive image pair for baseline SimSiam models.

## 2.3  LenSiam and baseline SimSiam

We utilize the LenSiam SSL pipeline shown in Figure 1 (a) and (b) for training on simulated paired images generated as described in Sections 2.1 and 2.2. Except for the augmentation, the remaining components of the model are kept consistent with those of the original paper of SimSiam [11]. For both LenSiam and the baseline SimSiam pipelines, we employ the ResNet101 model from the TORCHVISION library [31] as the backbone encoder.

For training LenSiam and the baseline SimSiam models, the loss for optimization combines symmetrized loss setting on representation $z$ and $p$. The term $z$ is the direct output of the encoder, and $p$ is the output of the predictor. We pass the lens images $(x_1, x_2)$ into the Siamese network twice but in a different order to obtain $(z_1, z_2)$ and $(p_1, p_2)$. The loss function is $\mathcal{L} = \frac{1}{2}\mathcal{D}(p_1, z_2) + \frac{1}{2}\mathcal{D}(p_2, z_1)$,

where $\mathcal{D}(p, z) = -\left[\frac{p}{\|p\|_2} \cdot \frac{z}{\|z\|_2}\right]$ defines the negative cosine similarity of $(p, z)$ and $\|\cdot\|_2$ is the $\ell_2$-norm. We adopt the `stopgrad` operation on $z$ and use `SGD` as our optimizer for training. The initial learning rate is set to 0.03 to optimize our training process.

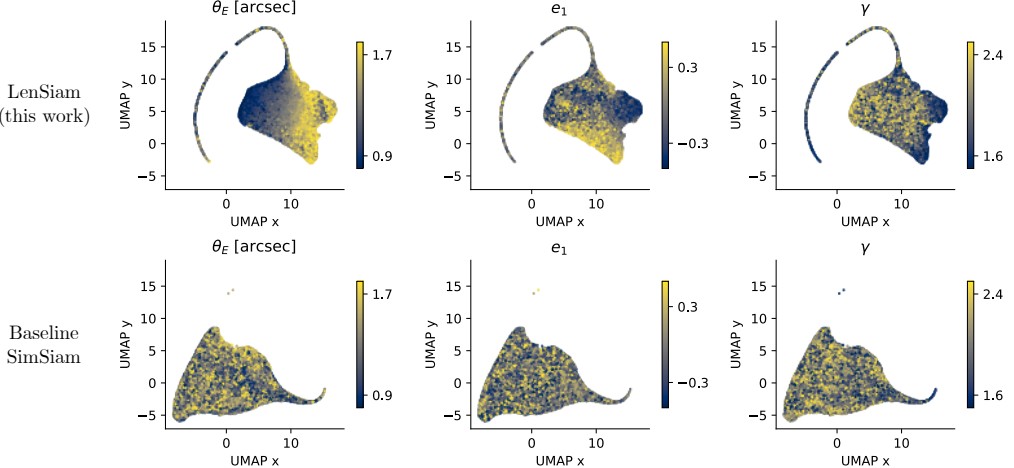

Figure 2: The UMAPs are color-coded by the Einstein radius $\theta_\mathrm{E}$, the ellipticity $e_1$, and the radial power-law slope $\gamma$ from the left to right columns. The top row is the UMAPs for LenSiam while the bottom row is the UMAPs for the baseline SimSiam.

## 3 The Learned Lens Image Representations

Here, we investigate the lens image representations learned from our best-trained LenSiam ResNet101 encoder with a loss of $-0.92$ and our best-trained SimSiam ResNet101 encoder with a loss of $-0.94$. We fit the Uniform Manifold Approximation and Projection (UMAP) [32] of each ResNet101 with its corresponding SSL 100,000 paired lens images. UMAP is a commonly used visualization method to understand high-dimensional representations by mapping them on the 2-dimensional UMAP space.

Figure 2 shows the UMAPs color-coded by lensing parameters the Einstein radius $\theta_\mathrm{E}$, the ellipticity $e_1$, and the radial power-law slope $\gamma$ for LenSiam (top panel) and the baseline SimSiam (bottom panel). The nonuniform distributions on the LenSiam UMAPs indicate that its backbone ResNet101 trained by the LenSiam SSL process does learn some key parameters such as $\theta_\mathrm{E}$, $e_1$, and $\gamma$, even though it has *NEVER* seen the true parameters during the entire training process. On the other side, the ResNet101 trained by the original SimSiam SSL does not learn them well given their relatively stochastic distributions on the UMAPs. We note that the UMAPs of the other parameters $e_2, \theta_1, \theta_2, \gamma_1, \gamma_2$ are roughly uniformly distributed for both LenSiam and baseline SimSiam models and those UMAPs are not shown in Figure 2 simply due to the limited space.

While the lensing parameters have been completely absent during the SSL training process, the learned representations of LenSiam are still capable of capturing lensing parameters whereas baseline SimSiam cannot. We believe this result is valuable from several perspectives. One, SSL has the ability to advance the lensing science by providing useful representation learning. Two, the lens augmentation with LenSiam is way more powerful compared to the default image augmentation and can be used for other Siamese Networks not limited to the SimSiam architecture. Finally, the lens image representations learned from our LenSiam process have the potential to improve downstream lensing tasks even if the data size is small in reality.

As an exploration, we experiment both our LenSiam and SimSiam learned representations with a downstream regression task as a proof of concept. We finetune the model to estimate the Einstein radius with the Lens challenge dataset, which simulated Euclid-like observations for strong lensing [33]. To simulate the scarcity of real strong lensing data, we select a sub-sample of 1,000 images as the training set and 1,000 images as the test set. With LenSiam pre-train models, we reach $0.586$ in $R^2$ compared with baseline SimSiam models $0.426$ and supervised-only models (the ResNet101 models pre-trained on ImageNet-1k) $0.360$ on Einstein radius. We find that the LenSiam pretraining does

help downstream regression task, hence shedding light on using the pretraining on ML-based strong lensing parameter estimation [7, 9]. We are planning to study the effectiveness of the pre-trained model on real lensing data as well as uncertainty estimation in future work.

## 4    Conclusion

In summary, we introduce LenSiam as a valuable approach to representation learning on lens images. It leverages lens augmentation for building good representation without any labels provided and adding to the performance of downstream tasks. This makes LenSiam an appealing choice for pretraining for ML-based lens image analysis. In future work, we plan to investigate LenSiam with real data and try different encoders (e.g., ViT). We believe self-supervised learning techniques like LenSiam will be beneficial for the strong lensing community.

## Acknowledgments and Disclosure of Funding

The authors thank the referees for their useful feedback and the lenstronomy community for making the gravitational lensing simulation available. Po-Wen Chang was supported by NSF Grant No. PHY-2310018 to John Beacom. Joshua Fagin and James Hung-Hsu Chan acknowledge support from the Schmidt Futures program. Joshua Yao-Yu Lin thanks Siddharth Mishra-Sharma for the useful discussion. The computational resources of this work were provided by the Ohio Supercomputer Center [34].

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
