# OpenReview forum: "LenSiam: Self-Supervised Learning on Strong Gravitational Lens Images"
_NeurIPS.cc/2023/Workshop/AI4Science — NeurIPS2023-AI4Science Poster_

### Official Review · Reviewer_LMze · 2023-10-09
**Good submission that analyzes representation learning for gravitational lens images**

**Rating:** 8
**Confidence:** 4

**Review:**

SUMMARY:

This submission applies representation learning to strong gravitational lens images. The author(s) propose(s) LenSiam architecture which is based on SimSiam but uses an augmentation technique that is specific to Lens images. The LenSiam method is evaluated by inspecting the UMAP embeddings of the learned features and another downstream regression task.

STRENGTHS:
- The paper is well-structured and reading it was fun. The author(s) seem(s) to have spend a lot of time to polish this submission. Figures are of high quality.
- The proposed ideas are well motivated.
- Necessary backgrounds are introduced. I have learned a lot while reading this paper.
- The experiments are well-executed and the results are convincing.


WEAKNESSES:
- The citation style is somewhat unusual (e.g., "Ref. [15] shows that SSL...") but that might be common in other research areas.
- Since SimSiam is not a contrastive method (as it does not use negative sample pairs), I would slightly change the sentence: "SimSiam is more computationally affordable than other contrastive learning methods". (I think dropping "other" already fixes this.)
- l. 124 says "those UMAPs are not shown in Figure 2 simply due to the limited space." However, the submission length for the AI4Science workshop is 4-8 pages. So there is enough space to show these figures...

CONCLUSION:

Overall, I think that this submission would make a nice contribution to this year's AI4Science workshop. I highly recommend to accept it.

---

### Meta-Review · Area_Chair_BmYV · 2023-10-27

**Recommendation:** Accept (Poster)
**Confidence:** 5

**Metareview:**

The author(s) introduce a self-supervised representation learning designed for strong gravitational lens images. They propose the augmentation technique that preserves lens properties, such as the Einstein radius. Additionally, they employ SimSiam, a method that only utilizes positive pairs to circumvent potential ambiguity in defining negative pairs.

This paper is well organized and provides the necessary background information in this domain. They provide a reasonable explanation for the challenges that existing baseline augmentation techniques can result in ambiguous positive views and empirically demonstrate these challenges. This aligns exceptionally well with the interests of this workshop. However, I concern that more extensive experiments with numerical comparisons are needed to strengthen this claim and it will be better if they also experimentally show that methods like SimCLR or BGRL lead to significantly poor results. Furthermore, as pointed out in the review, it's important to correct the description of SimSiam as a non-contrastive method. Moreover, since there is available space, it would be beneficial to include additional UMAP visualizations and other content.

I also suggest considering research directions to apply methods like AFGRL, which obtains superior representations without the necessity for augmentation. This could be a valuable approach to address the challenges outlined in your study.

While it is regrettable that only 4 out of the available 8 pages have been filled, I recommend accepting the paper due to the interesting nature of the proposed research.